# Relationship between Abnormal Placenta and Obstetric Outcomes: A Meta-Analysis

**DOI:** 10.3390/biomedicines11061522

**Published:** 2023-05-25

**Authors:** Shinya Matsuzaki, Yutaka Ueda, Satoko Matsuzaki, Hitomi Sakaguchi, Mamoru Kakuda, Misooja Lee, Yuki Takemoto, Harue Hayashida, Michihide Maeda, Reisa Kakubari, Tsuyoshi Hisa, Seiji Mabuchi, Shoji Kamiura

**Affiliations:** 1Department of Gynecology, Osaka International Cancer Institute, Osaka 541-8567, Japan; 2Department of Obstetrics and Gynecology, Osaka University Graduate School of Medicine, Osaka 565-0871, Japan; 3Department of Obstetrics and Gynecology, Osaka General Medical Center, Osaka 558-8558, Japan; 4Department of Forensic Medicine, School of Medicine, Kindai University, Osaka 589-8511, Japan

**Keywords:** abnormal placenta, circumvallate placenta, bilobed placenta, succenturiate lobe, placenta membranacea, multilobed placenta, systematic review

## Abstract

The placenta has several crucial physiological functions that help maintain a normal pregnancy. Although approximately 2–4% of pregnancies are complicated by abnormal placentas, obstetric outcomes remain understudied. This study aimed to determine the outcomes and prevalence of patients with abnormal placentas by conducting a systematic review of 48 studies published between 1974 and 2022. The cumulative prevalence of circumvallate placenta, succenturiate placenta, multilobed placenta, and placenta membranacea were 1.2%, 1.0%, 0.2%, and 0.004%, respectively. Pregnancies with a circumvallate placenta were associated with an increased rate of emergent cesarean delivery, preterm birth (PTB), and placental abruption compared to those without a circumvallate placenta. The succenturiate lobe of the placenta was associated with a higher rate of emergent cesarean delivery, whereas comparative results were observed in terms of PTB, placental abruption, and placenta previa in comparison to those without a succenturiate lobe of the placenta. A comparator study that examined the outcomes of multilobed placentas found that this data is usually unavailable. Patient-level analysis (*n* = 15) showed high-rates of abortion (40%), placenta accreta spectrum (40%), and a low term delivery rate (13.3%) in women with placenta membranacea. Although the current evidence is insufficient to draw a robust conclusion, abnormal placentas should be recognized as a high-risk factor for adverse outcomes during pregnancy.

## 1. Introduction

Placenta and fetal lineages spatially diverge in the initial days of embryogenesis, and the placenta develops into a single circular or oval-shaped organ during early to middle pregnancy [1,2,3,4]. The placenta has immune, endocrine, transportation, and physiological roles in maintaining normal pregnancy [5,6,7,8]. Placental development is essential for normal fetal growth. However, it is a complicated process, and the placenta may sometimes vary in shape, location, and size. Due to its critical function in circulating blood and oxygen, variations during pregnancy and placental variants (abnormal placenta) can sometimes result in complications.

Abnormal placenta can be categorized into circumvallate placenta, succenturiate placenta, multilobed placenta (bilobed placenta, placenta bilobate, bipartite placenta, and placenta duplex), placenta fenestrate, placenta membranacea, and ring-shaped placenta [4]. The incidence of abnormal placenta has been reported to range between 2 and 4%. Previous studies have reported placental variants to be a possible risk factor for adverse obstetric outcomes such as preterm birth (PTB) and placental abruption [9,10,11,12,13].

Although pregnancy complicated by an abnormal placenta is not rare, the association between an abnormal placenta and obstetric outcomes remains understudied. Moreover, a systematic review to examine the effects of abnormal placenta on obstetric outcomes has not yet been conducted, which would also investigate the prevalence of abnormal placenta. Furthermore, robust and comprehensive data describing obstetric and delivery outcomes in women with an abnormal placenta remain limited. Therefore, we conducted a systematic review to examine the incidence of abnormal placentas and related obstetric and delivery outcomes.

## 2. Materials and Methods

### 2.1. Ethical Considerations

The Institutional Review Board of Osaka International Cancer Institution exempted the need for ethical clearance for the present systematic review, as this study used publicly available and unidentified data. Unpublished patient data were not used in this study.

### 2.2. Systematic Literature Review Approach

This systematic review has been registered in the International Prospective Register of Systematic Reviews (PROSPERO): registration ID CRD42022336187. The original protocol aimed to examine the effect of abnormal cord insertion and placenta placement on obstetric and delivery outcomes. However, the current study excluded women with abnormal cord insertion due to excessive data found in one study. A systematic search of the literature regarding abnormal placentas was performed according to a modified protocol. The primary objective was to examine the effect of abnormal placentas on obstetric outcomes and also determine the incidence of abnormal placentas during pregnancy. The secondary objective was to explore the delivery outcomes of women with abnormal placentas.

### 2.3. Eligibility Criteria, Information Sources, and Search Strategy

A systematic literature search according to the Preferred Reporting Items for Systematic Reviews and Meta-Analyses guidelines [14] was performed on 30 April 2022, using three electronic databases (PubMed, Scopus, and the Cochrane Central Register of Controlled Trials). The search keywords used to identify relevant studies are listed in Appendix A. If Medical Subject Heading (MeSH) terms were available, these keywords were used in the PubMed and Cochrane databases to identify articles regarding abnormal placentas (Appendix A). 

Previous studies on abnormal placenta were screened by scrutinizing the titles, abstracts, and full texts of candidate published works, as previously described, with some modifications [15,16]. All the titles and abstracts were reviewed by Shinya Matsuzaki and Misooja Lee. As in our previous study [17], the type of study was defined according to the following criteria: (*i*) case report (1 or 2), (*ii*) case series (3–10), and (*iii*) original articles (11≥).

### 2.4. Study Selection

Studies fulfilling the following inclusion criteria were incorporated in the current study: (*i*) a comparative study (experimental group [pregnant women with abnormal placenta] vs. a control group [pregnant women without abnormal placenta]); (*ii*) studies wherein the incidence of abnormal placenta was clarified; (*iii*) studies where obstetric outcomes in women with abnormal placenta were examined; (*iv*) a case report or case series that reported a woman or women with abnormal placenta.

The following studies were excluded: (*i*) studies wherein the number of women with abnormal placentas was unclear; (*ii*) studies with missing or unavailable information regarding the outcomes of interest (obstetric outcomes: rate of PTB, fetal growth restriction [FGR], placenta abruption, intrauterine fetal death [IUFD], and placenta accreta spectrum [PAS]; delivery outcome: rate of cesarean delivery; incidence: incidence of abnormal placenta) in women with abnormal placenta; (*iii*) studies wherein specific outcomes of each type of abnormal placenta patients were unavailable, (*iv*) articles not written in English; (*v*) conference abstracts, narrative reviews, systematic reviews, and meta-analyses.

### 2.5. Data Extraction

Data were extracted, and tables were prepared by Shinya Matsuzaki using Excel 2021 (Microsoft, Redmond, WA, USA) and double-checked by Shinya Matsuzaki and Misooja Lee. The following information was documented: publication year, study location, leading author’s name, number of pregnant women, and outcomes of interest. 

### 2.6. Analysis of Outcome Measures and Assessment of Bias Risk

The outcomes of interest in this study were obstetric outcomes (rate of PTB, FGR, placenta abruption, IUFD, and PAS); incidence; and delivery outcomes (rate of cesarean delivery and emergent cesarean delivery). Since studies on placenta membranacea were scarce, patient-level analysis (combining information from case reports or case series) was performed.

The risk of bias in eligible studies was assessed using the risk of bias in non-randomized studies of interventions tool (ROBINS-I) in accordance with the previous studies [18,19,20,21].

### 2.7. Sensitivity Analysis

The association between abnormal placentas and assisted reproductive technology (ART) was examined during sensitivity analysis, the method of which has been described elsewhere [22]. Since the association between succenturiate lobe placenta and multilobed placenta was examined in our previous systematic review [17], only the relationship between circumvallate placenta and ART was determined in the current study. 

### 2.8. Meta-Analysis Plan

The odds ratios (ORs) of the risk of outcomes (rate of preterm birth, FGR, placental abruption, cesarean delivery, and emergent cesarean delivery) from the identified studies were calculated using the 95% confidence intervals (95% Cis) of the reported values. To calculate the incidence of abnormal placentas, the number of pregnant women with abnormal placenta was divided by the total number of pregnant women included in the analysis.

Study heterogeneity was assessed using the *I*^2^ statistic, which measures the percentage of total variation across the studies. According to the Cochrane Handbook for Systematic Reviews of Interventions (version 6.3), heterogeneity was assessed based on the value of *I*^2^, with some modifications from a previous study [23].

Meta-analysis was performed using RevMan, version 5.4.1 (Cochrane Collaboration, Copenhagen, Denmark). All images were created using RevMan 5.4.1. Images were edited using Adobe Illustrator CS5 (Adobe Systems, San Jose, CA, USA). Data from all outcomes were entered into RevMan 5.4.1, such that negative effect sizes or relative risks of <1 favored active intervention.

### 2.9. Statistical Analysis

Fisher’s exact test or the Chi-square test were used to examine differences in baseline demographics between the two groups. All statistical analyses were based on two-tailed hypotheses, and a *p*-value < 0.05 was considered statistically significant.

## 3. Results of the Systematic Review

### 3.1. Study Selection

The study selection scheme is illustrated in Figure 1. A total of 963 studies were screened. Of these, 48 studies comprising 1625 pregnant women with abnormal placenta and 167,566 pregnant women without abnormal placentas met the inclusion criteria of the current systematic review for descriptive analysis [3,9,10,11,12,13,24,25,26,27,28,29,30,31,32,33,34,35,36,37,38,39,40,41,42,43,44,45,46,47,48,49,50,51,52,53,54,55,56,57,58,59,60,61,62,63,64,65].

### 3.2. Study Characteristics

The metadata of the 48 eligible studies are shown in Appendix A [3,9,10,11,12,13,24,25,26,27,28,29,30,31,32,33,34,35,36,37,38,39,40,41,42,43,44,45,46,47,48,49,50,51,52,53,54,55,56,57,58,59,60,61,62,63,64,65]. The eligible studies were published between 1974 and 2022. All studies were retrospective in nature, and none of them were randomized controlled studies. Of the included studies (*n* = 48), 26 were case reports [3,25,27,29,32,35,36,37,38,39,40,41,42,43,47,48,49,50,51,53,56,57,58,63,64,65], 4 were case series [33,59,60,61], and the remaining 18 were original research articles [9,10,11,12,13,24,26,28,30,31,34,44,45,46,52,54,55,62]. The majority of the studies (approximately one-third) were from the United States of America (*n* = 15, 31.3%) [10,24,27,30,37,42,48,50,52,54,55,56,57,58,63], followed by Japan (*n* = 11, 22.9%) [9,11,12,29,39,44,45,46,49,51,60], Europe (*n* = 8, 16.7%) [32,33,41,43,53,59,61,62], and others [3,13,25,26,28,31,34,35,36,38,40,47,64,65].

Data regarding circumvallate placenta were available in 18 studies [9,10,11,12,24,26,28,29,30,31,36,44,45,52,54,55,57,62]; 20 studies [9,13,24,27,28,30,31,35,37,40,41,43,44,46,47,49,51,60,61,63] included data regarding succenturiate lobe placenta; and 10 studies [3,9,24,28,32,33,38,39,48,50] had data regarding multilobed placenta. Eight studies [34,42,53,56,58,59,64,65] examined obstetric outcomes in women with other types of abnormal placenta.

Six studies were comparator studies in the context of circumvallate placenta [11,12,26,44,45,62], four researched succenturiate lobe placenta, and no comparator study was available on multilobed placenta.

### 3.3. Risk of Bias of Included Studies

Risk of bias assessment for comparative studies (*n* = 8) demonstrated a possible severe bias (low quality) in all studies (Appendix A).

## 4. Results of the Meta-Analysis

### 4.1. Circumvallate Placenta

#### 4.1.1. Study Characteristics

Data regarding circumvallate placenta were reported in 18 studies [9,10,11,12,24,26,28,29,30,31,36,44,45,52,54,55,57,62]. Of these (*n* = 18), 15 studies were original research articles and 3 were case reports (Appendix A). These studies included a total of 946 women with a circumvallate placenta. Among the original articles (*n* = 15), 6 examined obstetric outcomes [11,12,26,44,45,62] and 1 was excluded [45] due to overlapping data with another study [12] (Table 1). The association between ART and the rate of circumvallate placenta was available in six studies [11,12,24,28,30,31] (Table 1). 

#### 4.1.2. Primary Outcome: Obstetric Outcomes

To examine the obstetric outcomes in women with a circumvallate placenta, the ORs of PTB, FGR, and placental abruption were calculated. Four studies examined the risk of PTB in women with a circumvallate placenta [11,12,26,62], four determined FGR [11,12,26,62], and three investigated placental abruption (Table 2). 

In the pooled analysis of four studies with random effects analysis due to considerable heterogeneity, the rate of PTB in women with circumvallate placenta was significantly higher (OR 6.56, 95%CI 2.44–17.63; heterogeneity: *p* < 0.01, *I*^2^ = 89%) compared to those without circumvallate placenta (Table 2, Figure 2). In the context of FGR, in the unadjusted pooled analysis using random effects analysis (*n* = 4), women with a circumvallate placenta had a significantly higher incidence of FGR than women without a circumvallate placenta (OR 3.53, 95%CI 1.63–7.62; heterogeneity: *p* < 0.01, *I*^2^ = 81%). Considering the lack of heterogeneity, we conducted a fixed effects analysis in the unadjusted pooled analysis to determine the risk of placental abruption in women with circumvallate placentas. In this analysis, women with circumvallate placenta were associated with a significantly higher risk of placental abruption (OR 10.65, 95% CI 6.16–18.41; heterogeneity: *p* = 0.48, *I^2^* = 0%) compared to those without a circumvallate placenta. 

#### 4.1.3. Co-Primary Outcomes: Incidence

The incidence of circumvallate placentas was examined in a total of 12 studies [9,10,11,12,24,28,31,44,45,52,55,62]. Of these, four studies were excluded from the analysis due to the overlapping study periods [12,28,44,45]. One study was excluded from the analysis since women with circummarginate placenta were not excluded [30]. The reported prevalence of circumvallate placenta ranged between 0.20 and 4.30%, and the cumulative rate of circumvallate placenta was 1.2% (Table 1: 349/29,863 cases).

#### 4.1.4. Secondary Outcome: Delivery Outcomes

Data regarding the rate of cesarean delivery in women with a circumvallate placenta were mentioned in three studies (Table 1) [11,12,26]. All the studies compared the rate of cesarean delivery between women with and without a circumvallate placenta. Considering the substantial heterogeneity, a pooled analysis with random effects was conducted to estimate the OR of cesarean delivery. The unadjusted pooled analysis (*n* = 3) demonstrated that women with a circumvallate placenta had a similar rate of cesarean delivery (OR 1.34, 95% CI 0.55–3.30; heterogeneity: *p* < 0.01, *I*^2^ = 91%) as those without a circumvallate placenta (Figure 3).

Two studies stratified the type of cesarean delivery into elective and emergent cesarean deliveries [11,12]. In the context of elective cesarean delivery in women with a circumvallate placenta, the rate of elective cesarean delivery (OR 0.57, 95%CI 0.26–1.24; heterogeneity: *p* = 0.17, *I*^2^ = 47%) was similar to those without a circumvallate placenta in the unadjusted pooled analysis. In contrast, women with circumvallate placenta were more likely to have an emergent cesarean delivery (OR 3.63, 95% CI 2.70–4.90; heterogeneity: *p* < 0.01, *I*^2^ = 88%) compared to those without a circumvallate placenta.

Two studies mentioned postpartum hemorrhage (PPH) in women with a circumvallate placenta. Suzuki et al. [12] reported that 19 of 139 women (13.7%) had >500 mL bleeding at the time of delivery, and 2 (1.4%) had >1000 mL bleeding. The rate of PPH between women with and without a circumvallated placentas was similar. McCarthy et al. [54] reported that 1 in 12 women with circumvallate placenta experienced PPH after 8 weeks of delivery due to retained placenta.

#### 4.1.5. Association between ART Pregnancy and Circumvallate Placenta

The association between ART pregnancy and a circumvallate placenta was determined in six studies [11,12,24,28,30,31]. Of these, two studies compared the rate of circumvallate placenta between ART and non-ART pregnancy [11,12]. The other four studies included only ART pregnancy, and the rate of circumvallate placenta was compared as follows: preimplantation vs. non-preimplantation genetic testing [24], cleavage vs. blastocyst embryo transfers [28], male vs. female fetuses [31], and fresh embryo vs. frozen embryo transfers [30] (Table 3). Evidently, a similar rate of circumvallate placentas was observed between the experimental and control groups. 

### 4.2. Succenturiate Lobe Placenta

#### 4.2.1. Study Characteristics

The outcomes of succenturiate lobe placenta were mentioned in 20 studies (Appendix A) [9,13,24,27,28,30,31,35,37,40,41,43,44,46,47,49,51,60,61,63]. Of these (*n* = 20), 8 studies were original research articles [9,13,24,28,30,31,44,46], 2 were case series [60,61], and the remaining 10 were case reports [27,35,37,40,41,43,47,49,51,63]. These studies included a total of 653 women with a succenturiate lobe placenta.

#### 4.2.2. Primary and Secondary Outcomes: Obstetric and Delivery Outcomes

A total of eight original research articles that examined obstetric outcomes or the relationship between ART pregnancy and the succenturiate lobe placenta have been published (Table 4). Of these (*n* = 8), three examined the obstetric outcomes and rate of emergent cesarean delivery, PTB, placental abruption, and placenta previa in women with succenturiate lobe placenta. Two studies had overlapping data [44,46]. Therefore, the study by Suzuki et al. in 2010 was excluded from the examination of the risk of emergent cesarean delivery, PTB, and placental abruption [44]. 

The ORs of emergent cesarean delivery, PTB, placental abruption, and placenta previa in women with succenturiate lobe placenta were determined in comparison to those without succenturiate lobe placenta (Table 5). In the pooled analysis, women with succenturiate lobe placenta were more likely to have emergent cesarean delivery (fixed effect analysis: OR 2.37, 95% CI 1.83–3.07; heterogeneity: *p* = 0.24, *I*^2^ = 26%) compared to those without succenturiate lobe placenta. On the other hand, comparative results were observed for PTB (random effect analysis: OR 2.13, 95% CI 0.92–4.92; heterogeneity: *p* = 0.08, *I*^2^ = 68%), placental abruption (fixed effect analysis: OR 1.50, 95% CI 0.43–5.26; heterogeneity: *p* = 0.69, *I*^2^ = 0%), and placenta previa (fixed effect analysis: OR 2.05, 95% CI 0.91–4.64; heterogeneity: *p* = 0.42, *I*^2^ = 0%).

Two original research articles examined the risk of PPH in women with succenturiate lobe placentas. In a Japanese study reported by Suzuki et al. [46], the OR of PPH at the time of delivery was significantly higher in women with succenturiate lobe placenta (>500 mL: OR 5.56, 95%CI 3.09–10.00; >1000 mL: OR 5.22, 95% CI 2.04–13.35) compared to those without succenturiate lobe placenta. In a Chinese study reported by Ma et al. [13], the OR of PPH at delivery was similar (>500 mL: OR 1.45, 95% CI 0.99–2.14; >1000 mL: OR 0.93, 95% CI 0.30–2.90) between women with and without succenturiate lobe placenta. The pooled OR of risk of PPH in women with succenturiate lobe placenta was: >500 mL: OR 2.79, 95%CI 0.74–10.46; heterogeneity: *p* < 0.01, *I*^2^ = 93%; >1000 mL: OR 2.26, 95% CI 0.37–13.75; heterogeneity: *p* < 0.01, *I*^2^ = 93%.

#### 4.2.3. Co-Primary Outcome: Incidence

The prevalence of succenturiate lobes of the placenta was reported in eight studies [9,13,24,28,30,31,44,46]. Three studies were excluded due to overlapping study periods [28,44,46]. The incidence of succenturiate lobe placenta ranged between 0.7–1.9% [9,13,24,30,31]. The cumulative prevalence of succenturiate lobes of the placenta was 1.0% (Table 4: 497/47,621 cases).

### 4.3. Multilobed Placenta

#### 4.3.1. Study Characteristics

Ten studies included women with multilobed placentas [3,9,24,28,32,33,38,39,48,50]. These studies included a total of 66 women (Appendix A).

#### 4.3.2. Primary Outcome: Obstetric Outcomes

Of the 10 studies (*n* = 10), three were original research articles. On the other hand, none of the studies examined the obstetric outcomes of women with a multilobed placenta. The remaining seven studies reported 12 women with multilobed placentas. However, obstetric outcomes were difficult to examine due to the limited data and heterogeneity of the included cases.

#### 4.3.3. Co-primary Outcome: Incidence

The prevalence of multilobed placentas was reported in two studies [9,24,28,31], of which one was excluded from the analysis due to the overlapping data [28]. The reported prevalence ranged between 0.14 and 0.64% and the cumulative incidence of multilobed placenta was 0.23% (Appendix A: 43/18,333 cases).

#### 4.3.4. Secondary Outcome: Delivery Outcomes

Delivery outcomes, including the rate of cesarean delivery and PPH, in women with a multilobed placenta were not reported in previous studies.

### 4.4. Placenta Membranacea

#### 4.4.1. Study Characteristics

Seven studies reported women with placenta membranacea (Appendix A) [34,42,56,58,59,64,65]. Among these (*n* = 7), five were case reports [42,56,58,64,65], one was a case series [59], and one was an original research article [34]. 

#### 4.4.2. Primary Outcome: Obstetric Outcomes

Patient-level analysis was performed to examine obstetric outcomes in women with placenta membranacea (Table 6). A total of 15 patients with placenta membranacea were included in the study. Among them (*n* = 15), the rate of abortion was 40% (6/15) and the rate of preterm birth excluding women with intrauterine fetal death was 26.7% (4/15). Three women had intrauterine fetal death (20.0%), and 2 of 15 women (13.3%) had full-term deliveries. Although the definition of PAS has not been clarified, 6 of the 15 women (40%) had PAS.

#### 4.4.3. Co-Primary Outcome: Incidence

The prevalence of placenta membranacea was mentioned only in one Chinese study. In the present study, the prevalence of placenta membranacea was 0.004% (Table 6: 3/79,862 cases).

#### 4.4.4. Secondary Outcome: Delivery Outcomes

After excluding abortion cases, nine eligible women were included in the analysis to examine the rate of cesarean deliveries. Of them (*n* = 9), five (55.6%) had cesarean delivery and two (22.2%) experienced PPH. None of the women who had abortions experienced PPH.

## 5. Discussion

### 5.1. Principal Findings

The principal findings of this study were as follows: (*i*) circumvallate placenta was associated with a high risk of emergent CD, PTB, FGR, and placental abruption; (*ii*) succenturiate placenta may be a high-risk factor for emergent CD and PPH, whereas it may not be a risk factor for PTB, FGR, or placental abruption; (*iii*) obstetric and delivery outcomes in women with multilobed placentas were limited; (*iv*) the prevalence rates of circumvallate placenta, succenturiate placenta, multilobed placenta, and placenta membranacea were 1.2%, 1.0%, 0.2%, and 0.004%, respectively.

### 5.2. Strengths and Limitations

The strength of the current study is that, to the best of our knowledge, it may be the first systematic review focusing on the obstetric outcomes of abnormal placentas. Our study revealed that, although available data was limited, the risks of circumvallate placenta, succenturiate lobe of placenta, and placenta membranacea could be identified. However, outcomes in women with multilobed placenta remains understudied. Obstetric and delivery outcomes of abnormal placenta should be examined in future studies. 

Irrespective of the absence of large-scale studies, such as a nationwide study, our study revealed the incidence of each type of abnormal placenta. To the best of our knowledge, the current study is the first to examine the incidence of abnormal placenta. Thus, we believe that these data would be useful for clinicians. For more accurate data analysis, nationwide studies examining the prevalence of abnormal placenta are warranted.

Nevertheless, this study had some limitations. First, since all the included studies were retrospective in nature, unmeasured bias (potential sources of confounding factors in the study included varying indications of cesarean delivery as well as definitions of FGR and placental abruption across the studies, unmatched patient backgrounds, and a limited number of women with abnormal placenta) may exist. Further, the wide range of study duration in the current study (1974–2022) may have resulted in a bias in the analysis of PTB and FGR. In particular, different definitions of abnormal placenta among studies may have caused severe bias. Thus, this point should be recognized as a substantial limitation of this study. 

Second, the quality of the diagnosis of an abnormal placenta was unknown in the original research articles included in the analysis. Most authors did not clarify the definitions and diagnostic criteria for abnormal placentas. These are critical limitations, and readers need to be aware of these issues when interpreting the results of the current study. 

Third, women with abnormal placenta are more likely to have co-existence of other placental or cord abnormalities, such as PAS, velamentous cord insertion, and vasa previa [9,17,66,67]. In addition, most studies did not perform multivariate analysis to exclude confounding factors due to the limited number of abnormal placentas. The coexistence of abnormalities and confounding factors may have affected the results of the current study.

Fourth, we modified the protocol of the systematic review due to excessive data in one study, which had the potential to cause bias. During the literature search, we excluded studies whose abstracts were unavailable in the search engines. Consequently, old studies and letters were excluded, and this is noteworthy. 

Fifth, the current study may have had a publication bias that should be recognized. For instance, women without a poor prognosis due to an abnormal placenta and undiagnosed cases may not have been reported. 

Lastly, since the number of included studies was limited, this study adds limited new findings to the current evidence. Nevertheless, current systematic review increases the robustness of findings regarding the outcomes of abnormal placenta. We believe that this systematic review provides the current evidence of the effect of an abnormal placenta on obstetric and delivery outcomes for clinicians. 

### 5.3. Comparison with Existing Literature

#### 5.3.1. Primary Outcomes: Obstetric Outcomes in Abnormal Placenta

Circumvallate placenta was associated with increased rates of FGR, PTB, and placental abruption in the present study. These findings were consistent with those of the previous studies [11,12,26,44,45,62]. A retrospective study from Japan reported that vaginal bleeding during the second trimester (39.1% vs. 0.6%, *p* < 0.01) and preterm premature rupture of membranes (39.1% vs. 4.0%, *p* < 0.01) were significantly higher in women with a circumvallate placenta (*n* = 92) compared to those without a circumvallate placenta (*n* = 9057). These features may lead to increased rates of PTB and placental abruption. 

Since a circumvallate placenta is associated with a poor prognosis during pregnancy, a definite method for detecting this type of abnormal placenta is warranted. Suzuki et al. reported that women with placental thickness greater than 3.0 cm had circumvallate placenta in approximately 20% of deliveries and proposed that measurement of placental thickness would be useful for screening for circumvallate placenta [45].

Unlike circumvallate placenta, studies that examined the obstetric outcomes of succenturiate lobes of the placenta, multilobed placenta, and placenta membranacea are scarce because of their rarity. Although obstetric outcomes regarding these types of abnormal placenta remain understudied, particular attention should be paid to the presence of type II vasa previa in women with succenturiate lobes of the placenta and multilobed placenta [17]. In our analysis, the outcomes of placenta membranacea were extremely poor, and only 2 of 15 women had term delivery. Although available data are limited, the fact that women with placenta membranacea are at a high risk of PTB, abortion, IUFD, and PAS should be recognized, and appropriate measures should be taken to prevent the complications.

#### 5.3.2. Co-Primary Outcomes: Incidence of Abnormal Placenta

The present systematic review reported the prevalence of abnormal placenta, while no studies have examined the trends of abnormal placenta. Moreover, data regarding patient characteristics in women with abnormal placenta are scarce, and therefore, it is difficult to estimate the trends associated with abnormal placenta. To determine the trends of abnormal placenta, a future nationwide study is required.

#### 5.3.3. Secondary Outcomes: Delivery Outcomes

Our study found that women with a circumvallate placenta and succenturiate lobes of the placenta were more likely to undergo emergent cesarean delivery compared to those without an abnormal placenta. The unique feature of the present study is the stratification of cesarean delivery into elective and emergent cesarean delivery, which showed that emergent cesarean delivery was performed more commonly in women with circumvallate placenta and succenturiate lobes of the placenta, whereas elective cesarean delivery was not. 

According to our hypothesis, the reason for the increased rate of emergent cesarean delivery may be the increased rate of FGR due to placental insufficiency in women with circumvallate placentas [11]. Since FGR with placental insufficiency may be associated with an increased rate of emergent cesarean delivery, we believe that our hypothesis is correct [68,69,70,71]. With respect to the increased rate of emergent cesarean delivery in women with a succenturiate lobe of the placenta, the unprotected fetal vessels connecting the lobes may be associated with an increased rate of emergent cesarean delivery due to abnormal fetal heart monitoring [17].

The association between abnormal placenta and fetal anomalies, cognitive and cardiovascular development is a riveting topic. For instance, single umbilical artery has been associated with increased rate of fetal anomaly [72,73]. However, the present systematic review found that none of the studies focused on the relationship between abnormal placenta and fetal anomaly. Our study revealed that circumvallate placenta was associated with increased rate of FGR (OR 3.53, 95%CI 1.63–7.62), and FGR has been correlated with adverse effects on neurological [74,75,76] and cardiovascular development [77,78] of the fetus depending on its severity. Therefore, circumvallate placenta can potentially be associated with adverse effects on neurological and cardiovascular development. Further studies are required to examine the association of abnormal placenta with fetal anomalies and development. 

#### 5.3.4. Sensitivity Analysis: ART and Abnormal Placenta

The rate of occurrence of circumvallate placenta was found to be similar between ART and non-ART pregnancies in the present study, whereas ART pregnancy was associated with an increased rate of a succenturiate lobe of the placenta in previous studies [46,79]. The relationship between ART pregnancy and a multilobed placenta has not been determined [17]. Unlike a circumvallate placenta, ART pregnancy has the potential to increase the rate of succenturiate lobe of the placenta, and frozen embryo transfer has been associated with a further increase in the rate of succenturiate lobes of the placenta. While ART, especially frozen embryo transfer, is associated with increased incidence of PAS [80], no study has examined the rate of occurrence of PAS in ART pregnancy with succenturiate lobe of the placenta.

A circumvallate placenta is an abnormality in the shape of the placenta. Succenturiate lobes of the placenta and multilobed placenta may result from localized atrophic changes due to poor decidualization [46,81,82]. These differences in the development of each type of abnormal placenta may cause differences in their relationship with ART pregnancy.

## 6. Conclusions and Implications

### 6.1. Implications for Practice

An abnormal placenta may be associated with adverse obstetric outcomes such as an increased rate of emergent cesarean delivery, PTB, FGR, and placental abruption. However, it is important to recognize that the prognosis depends on the type of abnormal placenta. Therefore, the results of the present systematic review will help clinicians be more aware of these adverse outcomes that may occur in women with abnormal placentas. In particular, the OR of placental abruption in women with a circumvallate placenta was high (approximately 10), and clinicians need to comprehend this risk to improve maternal and neonatal outcomes.

### 6.2. Implications for Research

Although abnormal placenta are associated with adverse obstetric outcomes, the characteristics of patients with abnormal placenta are understudied. Further studies are required to assess and analyze the characteristics of women with abnormal placenta. Being aware of the risk factors in patients with characteristics of abnormal placenta may help in the diagnosis of an abnormal placenta during pregnancy. A nationwide study focusing mainly on the risk of an abnormal placenta may address this problem. Statistical analyses, such as multivariate analysis, propensity score matching, and inverse probability of treatment weighing, that were not performed in the studies included in this the present systematic review, would be helpful in excluding confounding factors. 

No study has examined the mechanism of abnormal placenta. Therefore, future studies focusing on the development of abnormal placenta may shed light on some interesting findings. These examinations may be helpful in identifying the risk of abnormal placenta. To promote basic research, future studies on the relationship between ART and abnormal placenta are warranted. 

## Figures and Tables

**Figure 1 biomedicines-11-01522-f001:**
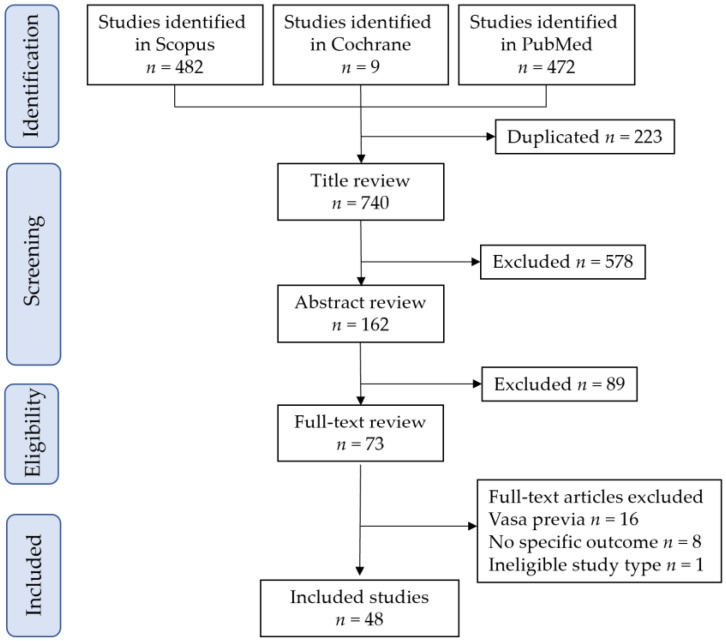
Study selection scheme of the current systematic review.

**Figure 2 biomedicines-11-01522-f002:**
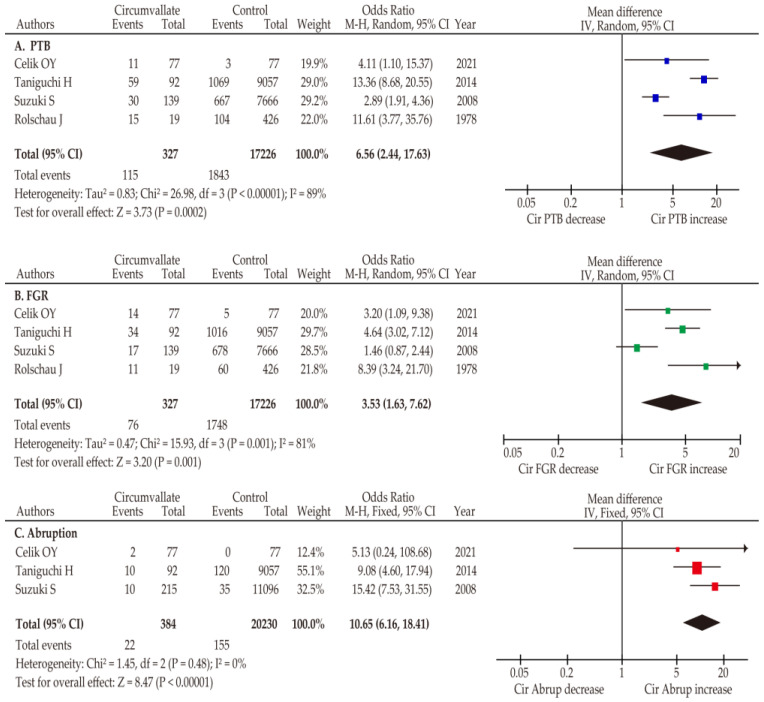
Meta-analysis of the effect of a circumvallate placenta on obstetric outcomes. The figure shows pooled ORs for (**A**) PTB (unadjusted) [11,12,26,62], (**B**) FGR (unadjusted) [11,12,26,62], and (**C**) placental abruption (unadjusted) [11,12,26] between women with and without at circumvallate placenta. Forest plots were ordered within the stratum by year of publication and relative weight (%) of the study. The size of colored boxes represents the weight of study and position is a point of the estimated odds ratio. Heterogeneity was considerable in the analysis of PTB and FGR ((**A**): *I*^2^ = 89%, (**B**): *I*^2^ = 81%), while there was none in the analysis of placental abruption ((**C**): *I*^2^ = 0%). Some values listed in the Figure may be slightly different from the original values since the calculations were done using RevMan ver. 5.4.1. Abbreviations: OR, odds ratio; CI, confidence interval; SE, standard error; PTB, preterm birth; FGR, fetal growth restriction.

**Figure 3 biomedicines-11-01522-f003:**
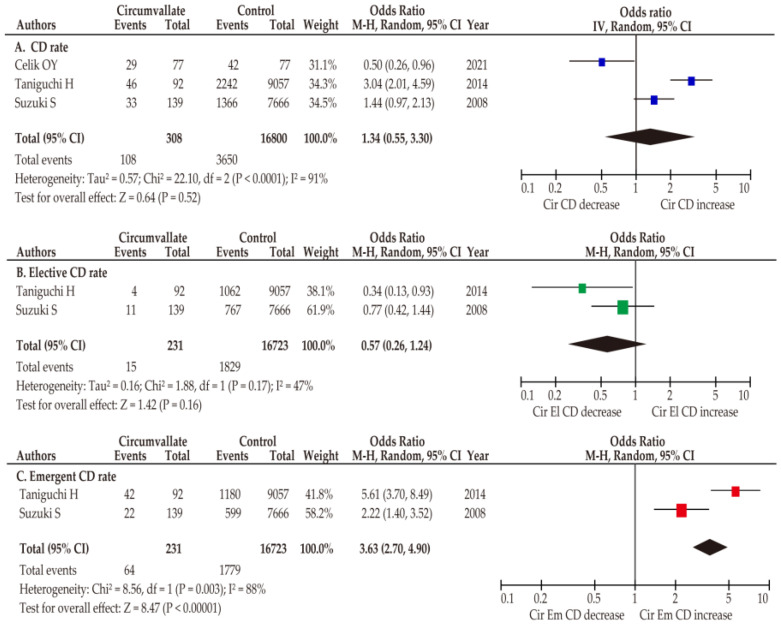
Meta-analysis of the effect of a circumvallate placenta on rate of cesarean delivery. The pooled odds ratios (ORs) for (**A**) cesarean delivery (unadjusted) [11,12,26], (**B**) elective cesarean delivery (unadjusted) [11,12], and (**C**) emergent cesarean delivery (unadjusted) [11,12] between women with and without a circumvallate placenta are shown. Forest plots were ordered within the stratum by year of publication and relative weight (%) of the study. The size of colored boxes represents the weight of study and position is a point of the estimated odds ratio. Considerable heterogeneity was observed in the analysis of cesarean delivery and emergent cesarean delivery ((**A**): *I*^2^ = 91%, (**C**): *I*^2^ = 88%), while the analysis of elective caesarian delivery showed moderate heterogeneity ((**B**): *I*^2^ = 47%). Some values in the Figure may be slightly different from the original values since the calculations were done using RevMan ver. 5.4.1. Abbreviations: CD, cesarean delivery; CI, confidence interval; SE, standard error.

**Table 1 biomedicines-11-01522-t001:** Summary of original research articles regarding the circumvallate placenta.

Author	Year	Total	*n*	Prevalence	ART	CD	ID	PTB	FGR ^&^	PA
Swanson K. [24]	2021	311	3	1.0%	3	--	--	--	--	--
Celik O.Y. [26] ^#^	2021	154	77	--	--	29	--	11	14	2
Volodarsky A. [28]	2021	677	8	1.2%	8	--	--	--	--	--
Volodarsky A. [31]	2020	1057	11	1.0%	11	--	--	--	--	--
Sacha C.R. [30] ^$^	2020	1140	62 *	--	62	--	--	--	--	--
Suzuki S. [9]	2015	16,965	217	1.3%	--	--	--	--	--	--
Taniguchi H. [11]	2014	9149	92	1.0%	4	46	3	59	--	10
Suzuki S. [44]	2010	11,311 ^†^	215	1.9%	--	--	--	--	--	10
Suzuki S. [45]	2008	722	11	1.5%	--	--	--	3	2	1
Suzuki S. [12]	2008	7930	139	1.8%	1	33	10	30	17	7
Ventolini G. [10]	2004	88	3	3.4%	--	--	--	--	--	--
Harris R.D. [52]	1997	62	1	1.6%	--	--	--	--	--	--
McCarthy J. [54]	1995	--	6	--	--	--	--	1	--	1
Sistrom C.L. [55]	1993	1784	3	0.2%	--	--	--	1	--	--
Rolschau J. [62]	1978	447	19	4.3%	--	--	--	6	11	--

^#^ This was a patient-matched study; thus, it was excluded in the analysis of the prevalence of circumvallate placenta. ^$^ This study was excluded from the analysis of the prevalence of circumvallate placenta since it included circummarginate placenta. * This study included circummarginate placenta. ^&^ This study included diagnoses after delivery. ^†^ This study included singleton pregnancies. Abbreviations: *n*, number of women with circumvallate placenta; --, not applicable; ART, Assisted reproductive technology; CD, cesarean delivery; ID, instrumental delivery; PTB, preterm birth; FGR, fetal growth restriction; and PA, placental abruption.

**Table 2 biomedicines-11-01522-t002:** Summary of circumvallate placenta comparative studies.

Author	Year	Total	Control	*n*	Em CD	PTB	FGR ^&^	PA
Celik OY [26]	2021	154	77	77	--	4.11 (1.10, 15.37)	3.20 (1.09, 9.38)	5.13 (0.24, 108.68)
Taniguchi H [11]	2014	9149	9057	92	5.61 (3.70–8.49)	13.36 (8.68, 20.55)	4.64 (3.02, 7.12)	9.08 (4.60, 17.94)
Suzuki S [44]	2010	11,311	11,096	215	--	--	--	15.42 (7.53, 31.55)
Suzuki S [45] *	2008	722	711	11	--	--	--	--
Suzuki S [12]	2008	7930	7666	139	2.22 (1.40–3.52)	2.89 (1.91, 4.36)	1.46 (0.87, 2.44)	--
Rolschau J [62]	1978	447	426	19	--	11.61 (3.77, 35.76)	8.39 (3.24, 21.70)	--

The odds ratios (ORs) and 95% confidence intervals (CIs) for each outcome are shown. * This study was excluded from the analysis due to the overlapping data. ^&^ including diagnosed after delivery. Abbreviations: *n*, number of women with circumvallate placenta; Em CD, emergent cesarean delivery; PTB, preterm birth; FGR, fetal growth restriction; PA, placental abruption; and --, not applicable.

**Table 3 biomedicines-11-01522-t003:** Association between ART and the circumvallate placenta.

Author	Year	Total	*n*	Exp.	Cont.	Exp. vs. Cont.	OR (95%CI)
Swanson K. [24]	2021	311	3	1/158	2/153	PGT vs non-PGT	0.48 (0.04, 5.36)
Volodarsky A. [28]	2021	679	8	1/252	7/425	Cleavage vs Blastocyst	0.24 (0.03, 1.94)
Volodarsky A. [31]	2020	1057	11	5/527	6/530	ART: Male vs Female	0.84 (0.25, 2.76)
Sacha C.R. [30]	2020	1140	62 *	56/929	6/211	Fresh vs Frozen	2.19 (0.93, 5.16)
Taniguchi H. [11]	2014	9149	92	4/359	88/8790	ART vs Non-ART	1.11 (0.41, 3.05)
Suzuki S. [12]	2008	7930	139	1/102	101/7703	ART vs Non-ART	0.75 (0.10, 5.39)

The effect of ART or type of ART on the rate of circumvallate placentas was determined. The calculated odds ratios and 95% confidence intervals comparing the experimental and control groups are shown. * including women with circummarginate placenta. Abbreviations: *n*, number of women with circumvallate placenta; Exp., rate of circumvallate placenta in the experimental group; Cont., rate of circumvallate placenta in the control group; vs., versus; OR, odds ratio; CI, confidence interval.

**Table 4 biomedicines-11-01522-t004:** Summary of original research articles regarding the succenturiate lobe placenta.

Author	Year	Total	*n*	Prev.	ART	CD	PTB	FGR ^&^	PA	PP
Swanson K. [24]	2021	313	6	1.9%	6	--	--	--	--	--
Volodarsky A. [28]	2021	677	6	0.9%	6	--	--	--	--	--
Volodarsky A. [31]	2020	1057	13	1.2%	13	--	--	--	--	--
Sacha C.R. [30]	2020	1030	70	--	70	--	--	--	--	--
Ma J.S. [13]	2016	28,256	294	1.0%	15	130	45	23	2	4
Suzuki S. [9]	2015	16,965	114	0.7%	--	--	--	--	--	--
Suzuki S. [44]	2010	11,311	83	0.7%	--	--	--	--	--	2
Suzuki S. [46]	2008	7713	47	0.6%	4	12	5		0	*

^&^ including diagnosed after delivery. * Excluded due to overlapping data. Abbreviations: *n*, number of women with succenturiate lobe placenta; --, not applicable ART, Assisted reproductive technology; Prev., prevalence; CD, cesarean delivery; ID, instrumental delivery; PTB, preterm birth; FGR, fetal growth restriction; PA, placental abruption; and PP, placenta previa.

**Table 5 biomedicines-11-01522-t005:** Obstetric outcomes of succenturiate lobe placenta.

Author	Year	Total	*n*	Em CD	PTB	PA	PP
Ma J.S. [13]	2016	28,256	294	2.50 (1.91, 3.26)	2.95 (2.16, 4.04)	1.36 (0.34, 5.52)	1.71 (0.63, 4.62)
Suzuki S. [44]	2010	11,311	83	--	--	--	3.44 (0.83, 14.24)
Suzuki S. [46]	2008	7703	47	1.40 (0.55, 3.56)	1.23 (0.48, 3.11)	2.55 (0.15, 42.31)	--
Pooled				2.37 (1.83, 3.07)	2.13 (0.92, 4.92)	1.50 (0.43, 5.26)	2.05 (0.91, 4.64)

The odds ratios (ORs) and 95% confidence intervals (CIs) for each outcome are shown. Abbreviations: Em CD, emergent cesarean delivery; PTB, preterm birth; FGR, fetal growth restriction; PA, placental abruption.

**Table 6 biomedicines-11-01522-t006:** Obstetric outcomes in women with placenta membranacea.

Author	Year	Total	*n*	Age	Abortion	CD	PPH	FGR	PTB	PA	IUFD	PAS
Tang L. [34]	2019	79,862	3	24	--	Yes	--	--	--	--	--	Yes
				29	--	--	--	Yes	28 wk	--	Yes	Yes
				20	--	--	--	--	25 wk	--	Yes	--
Ravangard S.F. [42]	2013	--	1	35	--	Yes	--	--	25 wk	Yes	--	--
Dinh T.V. [56]	1992	--	1	34	--	Yes	Yes	--	--	--	--	Yes
Greenberg J.A. [58]	1991	--	1	33	--	Yes	Yes	--	32 wk	--	--	Yes
Wilkins B.S. [59]	1991	--	7	Unk	--	--	--	--	24 wk	--	--	--
				Unk	19 wk	--	--	--	--	--	--	Yes
				Unk	18 wk	--	--	--	--	--	--	--
				Unk	23 wk	--	--	--	--	--	--	--
				Unk	17 wk	--	--	--	--	--	--	--
				Unk	17 wk	--	--	--	--	--	--	Yes
				Unk	--	Yes	--	--	30 wk	--	--	--
Wladimiroff J.W. [64]	1976	--	1	28	--	--	--	--	26 wk	--	Yes	--
Mathews J. [65]	1974	--	1	19	20 wk	--	--	--	--	--	--	--

Abbreviations: *n*, number of women with placenta membranacea; CD, cesarean delivery; PPH, postpartum hemorrhage; PTB, preterm birth; FGR, fetal growth restriction; PA, placental abruption; IUFD, intrauterine fetal death; PAS, placenta accreta spectrum; Unk, unknown; wk, weeks of gestation; and --, not applicable.

## Data Availability

All the reports used in this study have been published in the literature.

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
