# Peer review of "Relationship between Abnormal Placenta and Obstetric Outcomes: A Meta-Analysis"

_biomedicines, 2023, doi:10.3390/biomedicines11061522_

Round 1

Reviewer 1 Report

The article studies through a meta-analysis of the literature, different types of placenta and their relationship on obstetric pathology. The knowledge about the fact already exists and unfortunately the work does not contribute to my understanding anything new on what is already known, I would not accept the study for publication as it does not contribute knowledge on what is already known.

Author Response

Reviewer #1 

The article studies through a meta-analysis of the literature, different types of placenta and their relationship on obstetric pathology.

Reply:

Thank you for your positive comment. The authors would like to thank the reviewer for his/her constructive critique to improve the manuscript. We have made every effort to address the issues raised and to respond to all the comments. Please, find next a detailed, point-by-point response to the reviewer's comments. We hope that our revisions would meet the reviewer’s expectations.

Reviewer #1, comment #1

The knowledge about the fact already exists and unfortunately the work does not contribute to my understanding anything new on what is already known, I would not accept the study for publication as it does not contribute knowledge on what is already known.

Reply:

Thank you for the comment. We agree with the reviewer’s opinion and have added this point as a limitation of our study as follows: Lastly, since the number of included studies was limited, this study adds limited new findings to the current evidence. Nevertheless, the current systematic review in-creases the robustness of findings regarding the outcomes of abnormal placenta. We believe that this systematic review provides the current evidence of the effect of abnormal placenta on obstetric and delivery outcomes for the clinicians (Lines 514-518).

Reviewer 2 Report

The authors put together all the relevant studies showing abnormal placenta development and the gestational outcomes. They have successfully covered most of the relevant studies. It might be interesting to see the development of those new borns as their age progresses. If there is any strong correlation between cognitive and cardiovascular development with abnormal placentation during gestation. Any comments from the authors on this topic?

Author Response

Reviewer #2

Reviewer #2, comment #1

The authors put together all the relevant studies showing abnormal placenta development and the gestational outcomes. They have successfully covered most of the relevant studies. It might be interesting to see the development of those new borns as their age progresses. If there is any strong correlation between cognitive and cardiovascular development with abnormal placentation during gestation. Any comments from the authors on this topic?

Reply:

Thank you for your insightful comment. We agree with the reviewer’s comment that the association between abnormal placenta and fetal development would be an interesting topic to examine. We attempted to examine this association. However, none of the published studies have focused on this topic. We have added the discussion on this topic in the revised manuscript as follows: The association between abnormal placenta and fetal anomaly, cognitive and cardiovascular development is a riveting topic. For instance, single umbilical artery has been associated with increased rate of fetal anomaly. However, the present systematic review found that none of the studies focused on the relationship between ab-normal placenta and fetal anomaly. Our study revealed that circumvallate placenta was associated with increased rate of FGR (OR 3.53, 95%CI 1.63–7.62), and FGR has been correlated with adverse effects on neurological and cardiovascular development of the fetus depending on its severity. Therefore, circumvallate placenta can potentially be associated with adverse effects on neurological and cardiovascular development. Further studies are required to examine the association of abnormal placenta with fetal anomalies and development (lines 573-583).

Reviewer 3 Report

Dear authors,

This manuscript is a specific and exhaustive meta-analysis about abnormal placenta and obstetric problems. The manuscript is well written and structured, the introduction provides sufficient information and includes relevant references, the cited references are relevant to the research, research design is appropriate, the results are clearly presented, and the conclusions are supported by the results. Only two minor comments:

-          The title would be more appropriate as “Meta-analysis of relationship between abnormal placenta and obstetric outcomes”, so the manuscript is a meta-analysis, but not a review.

-          The authors should explain with more precision the exclusion criteria used.

Author Response

Reviewer #3

Dear authors,

This manuscript is a specific and exhaustive meta-analysis about abnormal placenta and obstetric problems. The manuscript is well written and structured, the introduction provides sufficient information and includes relevant references, the cited references are relevant to the research, research design is appropriate, the results are clearly presented, and the conclusions are supported by the results. Only two minor comments:

Reviewer #3, comment #1

-          The title would be more appropriate as “Meta-analysis of relationship between abnormal placenta and obstetric outcomes”, so the manuscript is a meta-analysis, but not a review.

Reply: Title

We sincerely appreciate the reviewer’s positive comments. We have revised the title of the manuscript in accordance with the reviewer’s suggestion.

Reviewer #3, comment #2

-          The authors should explain with more precision the exclusion criteria used.

Reply:

Thank you for the helpful comment. As per the reviewer’s comment, we have revised the exclusion criteria of our study (Lines 109-118).

Reviewer 4 Report

This is very interesting systematic review and meta-analysis which may have important implications for everyday clinical work, because in some cases it may be helpful to determine the perinatal risk even in so called low risk pregnancies. I have nothing to add to the methodology and well written paper.

Author Response

Reviewer #4

This is very interesting systematic review and meta-analysis which may have important implications for everyday clinical work, because in some cases it may be helpful to determine the perinatal risk even in so called low risk pregnancies. I have nothing to add to the methodology and well written paper.

Reply:

We sincerely appreciate the reviewer’s positive comments. We trust that the revised manuscript will now be suitable for publication in Biomedicines.

Round 2

Reviewer 1 Report

After reading the paper I consider that my oppinion is the same as the first evaluation. I don´t consider the paper to be published in the magazine.  My arguments are the same.